# Hashcash Tree, a Data Structure to Mitigate Denial-of-Service Attacks

**Mario Alviano** 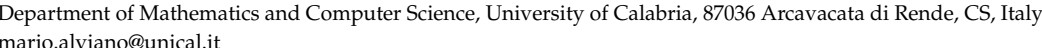

Department of Mathematics and Computer Science, University of Calabria, 87036 Arcavacata di Rende, CS, Italy; mario.alviano@unical.it

**Abstract:** Client puzzle protocols are widely adopted mechanisms for defending against resource exhaustion denial-of-service (DoS) attacks. Among the simplest puzzles used by such protocols, there are cryptographic challenges requiring the finding of hash values with some required properties. However, by the way hash functions are designed, predicting the difficulty of finding hash values with non-trivial properties is impossible. This is the main limitation of simple proof-of-work (PoW) algorithms, such as hashcash. We propose a new data structure combining hashcash and Merkle trees, also known as hash trees. In the proposed data structure, called hashcash tree, all hash values are required to start with a given number of zeros (as for hashcash), and hash values of internal nodes are obtained by hashing the hash values of child nodes (as for hash trees). The client is forced to compute all hash values, but only those in the path from a leaf to the root are required by the server to verify the proof of work. The proposed client puzzle is implemented and evaluated empirically to show that the difficulty of puzzles can be accurately controlled.

**Keywords:** security; partial hash collision; price functions; denial-of-service attacks; client puzzles; proof of work

## 1. Introduction

Denial-of-service (DoS) attacks are focused on making a resource (site, application, server) unavailable for the purpose it was designed for and represent a severe threat to the current Internet community [1]. DoS attacks cause significant losses [2,3] and motivate the research of sophisticated detection techniques [4–8]. Dwork and Noar [9] suggested the use of proof-of-work (PoW) schema to mitigate the proliferation of spam emails: a computation stamp is required to obtain a service; in the context of emails, the service can be the forwarding of a message. In general, PoW is a form of cryptographic proof in which one party (the prover) proves to others (the verifiers) that a certain amount of a specific computational effort has been expended; verifiers can subsequently confirm this expenditure with minimal effort on their part [10]. PoW schema are dissymmetric in favor of the verifier: the computation is moderately hard for the prover, while it is easy for a verifier to check a given solution. When PoW is applied to DoS mitigation, the prover role is played by a client aiming at accessing a service, and the verifier role is played by the server providing the required service.

PoW is often implemented by solving a cryptographic puzzle. The puzzle can be chosen by the sever, leading to *challenge–response protocols*, or self-imposed by the request of the client, leading to *solution–verification protocols*. In this article, protocols of these kinds are collectively called *client puzzle protocols*. Figure 1 shows the main steps of challenge–response protocols. A client, acting as prover, needs a service provided by a server, acting as verifier (step 1). The verifier generates a puzzle, with negligible effort, and challenges the client (steps 2–3). The client affords a moderately hard computation to solve the puzzle, and sends the solution to the server (steps 4–5). The server verifies the solution, again with negligible effort, and grants access to the requested service (steps 6–7). Figure 2 shows

the main steps of solution–verification protocols. The client generates a puzzle based on the requested service and computes a solution (steps 1–2). After that, the client sends the puzzle, its solution, and the request to the server (step 3). The server verifies that the puzzle was correctly generated and that the solution is valid (step 4). If both tests succeed, the request is processed (step 5).

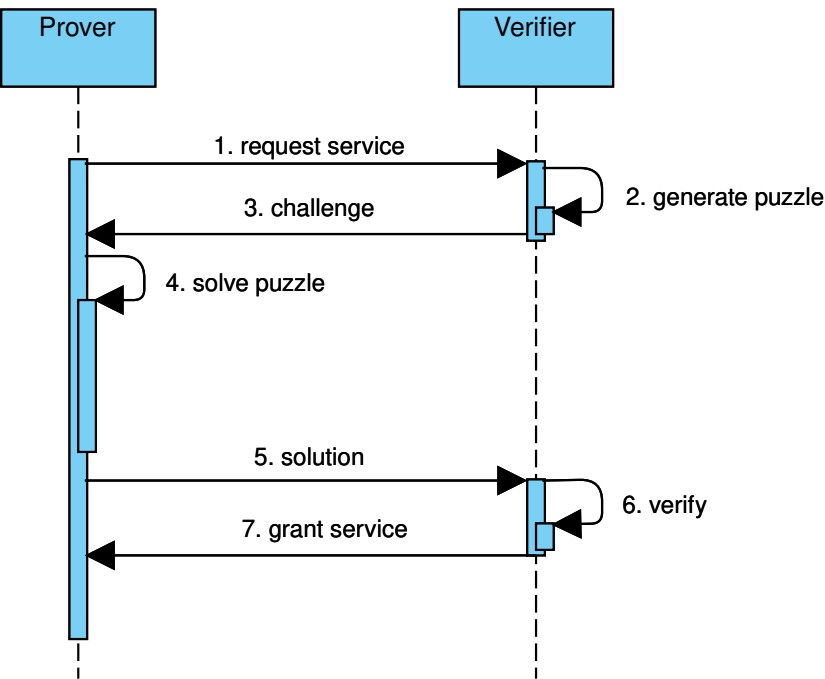

**Figure 1.** Sequence of events in a challenge–response protocol.

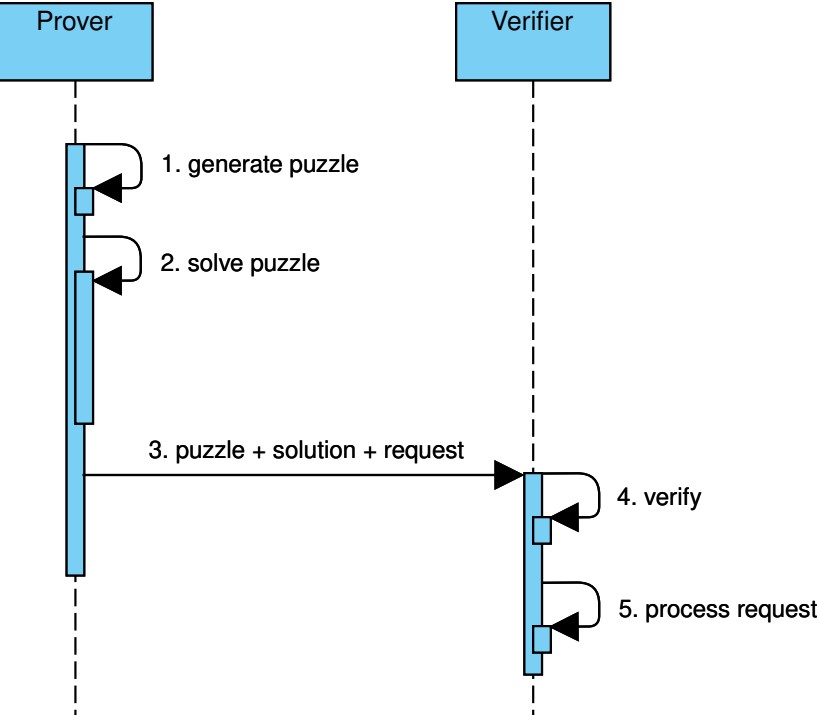

**Figure 2.** Sequence of events in a solution–verification protocol.

Hashcash [11] is a cryptographic hash-based PoW algorithm that can be used for defining both challenge–response protocols and solution–verification protocols. In a challenge–

response protocol based on hashcash, the verifier challenges the prover to find an extension of a given string whose hash value starts with a given number of zeros. As a PoW, the prover sends the extended string and its hash value, that is, the output of the hashcash algorithm. In a solution–verification protocol, the hashcash algorithm is applied to the service description, with a prefix length fixed in the protocol. Dissymmetry is given by the fact that the prover must try several extensions of the given string to find one with the required prefix, while the verifier needs only one hash value computation to verify the provided solution.

The main downside of hashcash is that the difficulty of the puzzle only depends on the length of the required all-zeros prefix. Adding a single zero to the prefix doubles the number of attempts that the prover must afford, and also the variance increases exponentially. A much better control on the difficulty of the puzzle was obtained by Coelho [12], who proposed a *solution–verification protocol based on hash trees*. In such trees, every leaf is labeled by the hash value of the leaf index concatenated with the service description. Every internal node is labeled by the hash value of the string obtained by concatenating child labels. Finally, some leaves are selected based on the hash value of the root. The prover constructs the tree and sends to the verifier the nodes in the paths from the selected nodes and their children (actually, the set of nodes is shrunk by removing nodes whose children belongs to the set; see Figure 3). This way, the verifier has sufficient data to verify that the puzzle was generated and solved correctly. While such a protocol succeeds in controlling the growth of the prover's effort, in practice it needs very large hash trees that must be either stored for the full computation or recomputed after the root hash is determined.

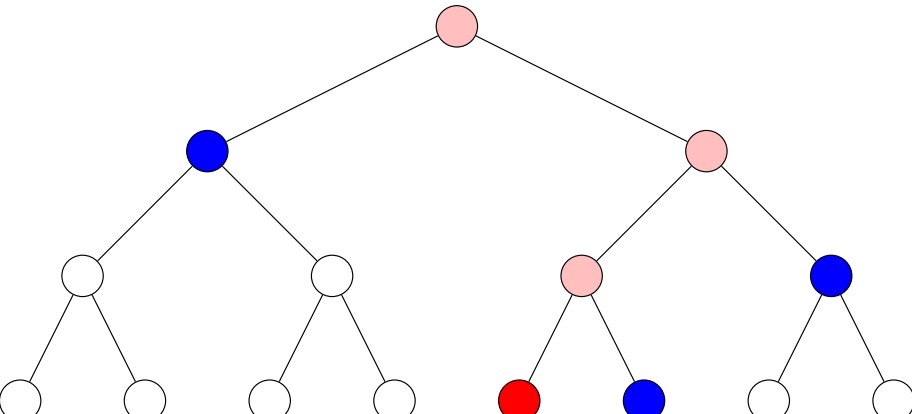

**Figure 3.** In a hash tree, to verify the hash value of the red node, the hash values of the blue nodes are used to reconstruct the hash value of the root. Pink nodes are those in the path from the selected node to root. Blue nodes are their children not included in the path.

In a nutshell, this article addresses the following research questions:

**RQ1** Is it possible to define a client puzzle protocol based on hashcash but with more controllable difficulty of puzzles?

**RQ2** Is it possible to define a client puzzle protocol based on hash trees but requiring smaller trees than those used by Coelho [12]?

**RQ3** How resistant to parallel computation is the proposed client puzzle protocol?

To answer the above questions, the challenge–response protocol is modified by splitting the challenge in two phases, as shown in Figure 4. In the first phase, the prover is challenged to solve a puzzle generated by the verifier (steps 2–3). After solving the puzzle (step 4), the prover sends to the verifier a *commitment* to conclude the first phase (step 5). In the second phase, the prover is asked to provide a proof of the solution, that is, sufficient data to verify that the prover has computed the committed solution (steps 6–9). In the proposed protocol, the puzzle consists of building a tree satisfying the following conditions: every leaf is labeled by the output of the hashcash algorithm applied on a verifier-provided

string concatenated to the leaf index; every internal node is labeled by the output of the hashcash algorithm applied on the verifier-provided string concatenated to the node index and the hash values of child nodes. This way, the difficulty of the construction of the tree can be controlled by varying the length of the all-zeros prefix and the number of nodes in the tree. Similarly to hash trees, the verification of a solution involves the selected node, those in the path from the selected node to the root, and their children. Given the combination of features from hashcash and hash trees, the proposed data structure is named *hashcash tree*.

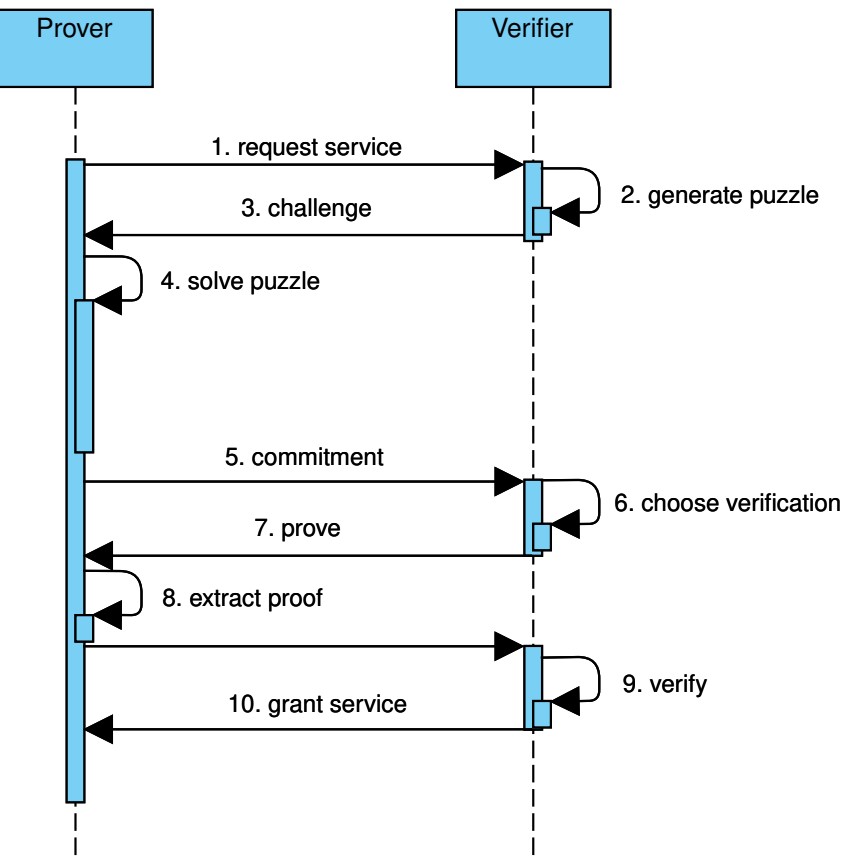

**Figure 4.** Sequence of events in a two-phases challenge–response protocol.

Hashcash trees and the two-phases challenge–response protocol relying on them are presented in Section 3, after introducing the required background in Section 2. Properties of the proposed client puzzle protocol are discussed in Section 4. In particular, the computational complexity is analyzed in terms of computed hash values for both the construction of hashcash trees and their verification. As for generating hashcash trees, the average number of computed hash values grows exponentially on the length of the all-zeros prefix and linearly on the number of nodes. Variance has a similar trend. It turns out that the grow can be controlled by fixing a relatively small length for the prefix and varying only the number of nodes depending on the workload of the verifier. As for the verification phase, the effort only depends (logarithmically) on the number of nodes. The construction of hashcash trees can be only partially parallelized, as the computation of a node label depends on the label of child nodes. An empirical evaluation of a proof-of-concept implementation of the protocol is reported in Section 5. The results confirm that the client puzzle protocol based on hashcash trees can be used in practice to challenge the prover in generating trees whose construction effort in terms of the number of computed hash values is controllable. Finally, related work is discussed in Section 6, and the article conclusion is summarized in Section 7.

In summary, the article answers **RQ1** and **RQ2** by introducing a new data structure combining hashcash and hash trees, namely hashcash trees. The computation of a single node of a hashcash tree is more expensive than a node in a hash tree because it requires solving a hashcash challenge. Thanks to such a more expensive computation, hashcash trees of relatively small size can be adopted by the proposed client puzzle protocol (**RQ2**). At the same time, solving several hashcash challenges of modest difficulty enables a fine-grained control on the difficulty of puzzles, which is not possible to achieve with a single hashcash challenge (**RQ1**). Finally, the fact that a node in a hashcash tree can be computed only after knowing the labels of child nodes provides a restricted form of parallel resistance (**RQ3**).

## 2. Background

This section introduces preliminary notions such as strings and hash functions (Section 2.1). Moreover, a client puzzle protocol based on the hashcash nondeterministic function is defined in Section 2.2. Finally, the main notation used for trees and the notion for hash trees are given in Section 2.3.

### 2.1. Hash Functions

The term *string* is used to refer binary strings, that is, elements in $\{0,1\}^*$; strings are also seen as sequences of bits. The *empty string* is denoted by $\epsilon$. The *prefix* of length $n \in \mathbb{N}$ of a string $s$ is the string consisting of the first $n$ bits of $s$ and is denoted by $prefix(s, n)$. The *concatenation* of two strings $s, s'$ is denoted by $s||s'$. For a string $s$ and a non-negative integer $x$, the notation $s||x$ is abused to denote the concatenation of $s$ with the binary string representation of $x$. The concatenation of a string $s$ with itself $n \in \mathbb{N}$ times is denoted by $s^n$ and defined inductively as follows: $s^0 = \epsilon$; $s^{n+1} = s^n||s$ for $n \geq 1$. The *i*-th element of a string $s$, and in general of a sequence or tuple, is denoted by $s|_i$.

**Example 1.** *The concatenation* $10101000||237$, *assuming an 8-bit representation of the integer* $237$, *is* $1010100011101101$. *The string* $(101)^3$ *is* $101101101$.

A *hash function* is any function mapping strings of arbitrary length to strings of fixed length. If $m \in \mathbb{N}^+$ is the fixed length, the signature of the hash function is

$$h : \{0,1\}^* \to \{0,1\}^m. \tag{1}$$

For a string $s \in \{0,1\}^*$, the value $h(s)$ returned by the hash function $h$ is called the *hash value* of $s$; the string $s$ is called *message*. Cryptographic hash functions additionally ensure the following properties:

(i)   The probability of a particular hash value for a message is $2^{-m}$;
(ii)  Finding a message that matches a given hash value is unfeasible (*preimage resistance*);
(iii) Finding a second message that matches the hash value of a given message is unfeasible (*second-preimage resistance*);
(iv)  Finding two different messages that yield the same hash value is unfeasible (*collision resistance*).

**Example 2.** *SHA-256 is a popular cryptographic hash function. It produces hash values of length 256. In Unix systems, the hexadecimal representation of the SHA-256 hash value of the string* `foo` *can be obtained as shown in Figure 5. Note that the produced hash value starts with* `0x2c = 00111100`, *that is, has an all-zero prefix of length 2.*

```
$ echo -n "foo" | sha256sum
2c26b46b68ffc68ff99b453c1d30413413422d706483bfa0f98a5e886266e7ae  -
```

**Figure 5.** Computing the SHA-256 hash value of the string "foo" in Unix-like systems.

### 2.2. A Client Puzzle Protocol Based on Hashcash

*Hashcash* is a (nondeterministic) function parameterized by a hash function $h$ and a positive integer $k$; it associates every input string $s \in \{0,1\}^*$ with any pair $\langle h(s||x), x \rangle$ such that $x \in \mathbb{N}$ and $\textit{prefix}(h(s||x), k) = 0^k$. Let $h_k(s)$ denote any valid output $\langle h(s||x), x \rangle$ of the hashcash function parameterized by $h$ and $k$. Hashcash essentially asks for finding partial hash collisions on the all-zeros prefix of length $k$, and the fastest algorithm for computing partial collisions is brute force [13]; see Algorithm 1. Assuming that the brute force algorithm increments $x$ starting from $x = 0$, finding a valid $\langle h(s||x'), x' \rangle$ output value requires the computation of $x' + 1$ hash values. In general, hashcash has *unbounded probability cost*, in the sense that theoretically the brute force algorithm can run forever, though the probability that a solution is not found decreases rapidly towards zero. On the other hand, verifying that $\langle h(s||x), x \rangle$ is a valid output requires the computation of one hash value, as shown in Algorithm 2.

---

**Algorithm 1:** HASHCASH($s, h, k$)

1 **for** $x \in \mathbb{N}$ **do**
2      *hash_value* $:= h(s||x)$;
3      **if** *prefix(hash_value, k)* $= 0^k$ **then**
4          **return** $\langle \textit{hash\_value}, x \rangle$;

---

**Algorithm 2:** HASHCASHVERIFY($\langle \textit{hash\_value}, x \rangle, s, h, k$)

1 **return** *prefix(hash_value, k)* $= 0^k$ **and** *hash_value* $= h(s||x)$;

---

**Example 3.** *Figure 6 shows a Bash script implementing Algorithm 1 and its execution for different lengths of the all-zeros prefix. Time measured on an Intel(R) Core(TM) i7-7600U CPU @ 2.80GHz. It can be observed that the difficulty of the problem does not increase linearly with the number of zeros in the demanded prefix.*

```
$ cat hcash.sh
input="$1"; prefix="$2"; x=0
while true; do
    string=`echo -n "$input||$x" | sha256sum`
    if [[ "$string" =~ ^"$prefix" ]]; then
        echo "hash = $string\n   x = $x"; break
    fi
    x=$((x + 1))
done
$ for p in 0 00 000 0000; do /bin/time -f "time = %Us" bash hcash.sh foo $p; done
hash = 0aa4f8fc0f89d777a93a6784d001b017e87226998ecfbaacd1d81718f160cd9a  -
   x = 3
time = 0.01s

hash = 00c679e2f3093d2d437c22ef2c35673e66829ca17f575eaba81699a549d23b7e  -
   x = 442
time = 0.62s

hash = 0009d063183b10c0c8d1906674aac59b2831b92f771e41ecdeb69ca1aaf0f701  -
   x = 479
time = 0.69s

hash = 000056ea4be004ebf657e89dd5f129bac5f199ff73d1dacafdfd0d440c0ac473  -
   x = 56358
time = 89.15s
```

**Figure 6.** A Bash implementation of hashcash and its execution for prefixes $0^k$ with $k \in [1..4]$.

A *client puzzle protocol* (CPP) involves two entities, namely a verifier $V$ and a prover $P$. $P$ needs to access some (computationally expensive) resource of $V$. $V$ challenges $P$ to

solve a puzzle before processing its request. A simple CPP based on hashcash comprises the following steps:

1.  The setup consists in $V$ generating and storing a master key $mk$. The master key is used to sign data so that $V$ can complete the protocol without storing any further data. (No signed data is extended in the protocol; no need for HMAC.)
2.  $P$ needs to send a request $req$ to $V$. To this aim, $P$ sends $Req := h(req)$ to $V$. (Here $req$ is considered unique. For example, $req$ can include a *nonce* chosen by $P$ and a timestamp, and $V$ can track $Req$ for all completed $req$ within the allotted time window. To simplify the presentation, such details are omitted from the discussion.)
3.  $V$ determines the *difficulty parameter* $k \in N^+$ based on its current workload, generates a timestamp $t$ by which the protocol must be completed, computes $s := h(mk||Req||k||t)$, and sends $\langle s, k, t \rangle$ to $P$. (Note that the hash value of the request is signed, not the request, so that the signed message has fixed length.)
4.  $P$ computes $S := h_k(s)$ and sends $\langle req, S, s, k, t, Req \rangle$ to $V$.
5.  $V$ checks all the following conditions: $s = h(mk||Req||k||t)$; $t$ is in the future; $S$ is valid, i.e., $S|_1 = h(s||S|_2)$ and $prefix(S|_1, k) = 0^k$; $Req = h(req)$. If all conditions are met, $req$ is processed.

The idea of a CPP is that a legit user is willing to waste a small amount of its computational resources in order to access a resource of $V$, while an attacker requires an unaffordable amount of computational resources to exhaust $V$ capabilities. The main downside of the above simple protocol is that the difficulty parameter $k$ does not give a fine-grained control on the amount of resources needed to solve the puzzle (no *determinable difficulty*). Moreover, the brute-force algorithm for hashcash has linear speedup (weak *parallel computation resistance*).

**Example 4.** *As shown in Example 3, for the message* foo *there is essentially no difference in increasing the prefix from* $0^8$ *to* $0^{12}$*, while increasing the prefix to* $0^{16}$ *makes the problem much more difficult. Also note that Algorithm 1 can be easily parallelized, and a GPU with 384 cores would solve the prefix* $0^{16}$ *by computing around 150 hash values per core in around 0.25 s.*

*2.3. Trees*

A *labeled binary tree* $T$ is either the empty set $\varnothing$ or a quadruple $\langle r, \ell, L, R \rangle$, where $r$ is the *root* node (of $T$), $\ell$ is the *label* (of $r$), and $L, R$ are labeled binary trees referred to as the *left child* and the *right child* (of both $T$ and $r$); $r$, $L$ and $R$ are respectively denoted by $root(T)$, $left(T)$ and $right(T)$. (In the following, the term *tree* is used to refer labeled binary trees.) Nodes of a tree are defined inductively: $nodes(\varnothing) := \varnothing$; $nodes(\langle r, \ell, L, R \rangle) := \{r\} \cup nodes(L) \cup nodes(R)$. When $L = R = \varnothing$, node $r$ is also called a *leaf*. Leaves of a tree are defined inductively: $leaves(\varnothing) := \varnothing$; $leaves(\langle r, \ell, \varnothing, \varnothing \rangle) := \{r\}$; $leaves(\langle r, \ell, L, R \rangle) := leaves(L) \cup leaves(R)$, if $L \neq \varnothing$ or $R \neq \varnothing$. An *internal node* of $T$ is any non-leaf node of $T$, that is, $internal(T) := nodes(T) \setminus leaves(T)$.

Let $v \in nodes(T)$ be a node of $T = \langle r, \ell, L, R \rangle$. The label of $v$ in $T$ is defined inductively: $label(v, \varnothing) := \epsilon$; $label(v, T) := \ell$ if $v = r$; $label(v, T) := label(v, L)$ if $v \in nodes(L)$; $label(v, T) := label(v, R)$ if $v \in nodes(R)$. The *level* of $v$ in $T$ is defined inductively: $level(v, T) := 1$ if $v = r$; $level(v, T) := 1 + level(v, L)$ if $v \in nodes(L)$; $level(v, T) := 1 + level(v, R)$ if $v \in nodes(R)$. A tree is *perfect* if all interior nodes have nonempty children and all leaves have the same level. (In the following, only perfect trees are considered.) The *height* of a tree is the level of its leaves. The *order* of $v$ in $T$ is defined via *breadth-first search* (BFS) or *level-order search*: $order(r, T) := 1$; for every $\langle v, \ell', L', R' \rangle$ occurring in $T$, $order(root(L'), T) := 2 \cdot order(v, T)$ and $order(root(R'), T) := 2 \cdot order(v, T) + 1$. Note that $T$ can be compactly represented by a one-based array $arr$ of size $|nodes(T)|$, having $arr[i] = label(v, T)$ whenever $order(v, T) = i$.

**Example 5.** *Let $T$ be the tree of height 4 shown in Figure 7. $T$ is such that $label(v, T) = order(v, T)$ for all $v \in nodes(T)$. Its array representation is $[1, 2, 3, 4, 5, 6, 7, 8, 9, 10, 11, 12, 13, 14, 15]$. The*

*leaves have level 4 and are the nodes with label from* 8 *to* 15. *Inner nodes have label from* 1 *to* 7. *The root has label* 1.

A *hash tree*, or *Merkle tree*, is a tree in which every leaf is labeled with the hash value of a data block, and every internal node is labeled with the hash value of the labels of its child nodes. Here a data block is any piece of information in a commitment scheme, that is, hidden data that cannot be changed. In fact, sharing the hash value associated with the root is sufficient to guarantee that no label of the hash tree is modified. If $T$ is the hash tree associated with a sequence $s_1, \ldots, s_{2^n}$ of data blocks ($n \in \mathbb{N}$), in order to verify that some $s_i$ ($i \in [1..2^n]$) was not modified, it is sufficient to check nodes in the path from the leaf with label $h(s_i)$ to the root. The check involves the labels of these nodes and their children; in formulas, if $v_o$ denotes the node of $T$ such that $oder(v_o, T) = o$, the labels involved in the verification of $s_i$ are $label(v_{\lfloor i \cdot 2^{-j} \rfloor}, T)$ and $label(v_{\lfloor i \cdot 2^{-j} \rfloor + 1}, T)$, for $j \in [0..n-1]$.

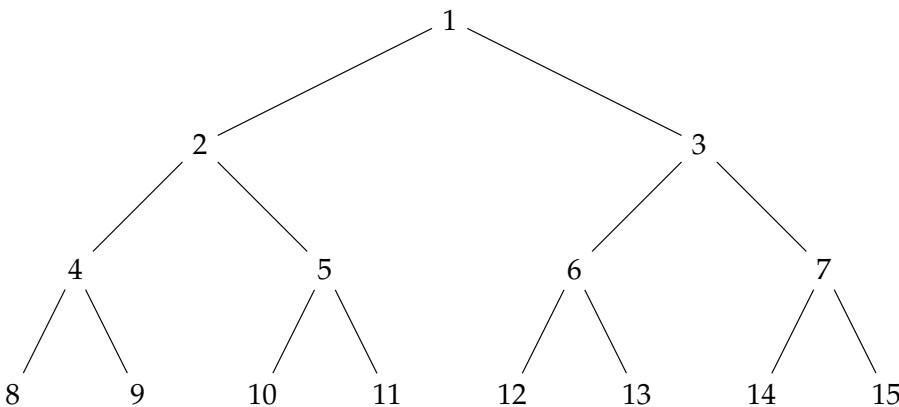

**Figure 7.** A perfect tree T whose nodes are labeled by their order.

**Example 6.** *Let $T$ be the hash tree shown in Figure 8. Let $v_o$ denote the node of $T$ such that $oder(v_o, T) = o$. To verify that $s_5$ was not modified, the nodes involved are $v_1, v_2, v_3, v_6, v_7, v_{12}, v_{13}$. In particular, the hash values that are recomputed are those associated with nodes from the leaf with label $h(s_5)$ to the root, i.e., those of $v_{12}, v_6, v_3$ and $v_1$. Essentially, to modify $s_5$ without changing the label of $v_1$ requires violating second-preimage resistance of $h$ at each level of $T$: find new labels for $v_2$ and $v_3$ matching the hash value in $v_1$; find new labels for $v_6$ and $v_7$ matching the new hash value in $v_3$; find a new label $\ell$ for $v_{13}$ such that $h(h(s_5')||\ell)$ matches the new hash value in $v_6$.*

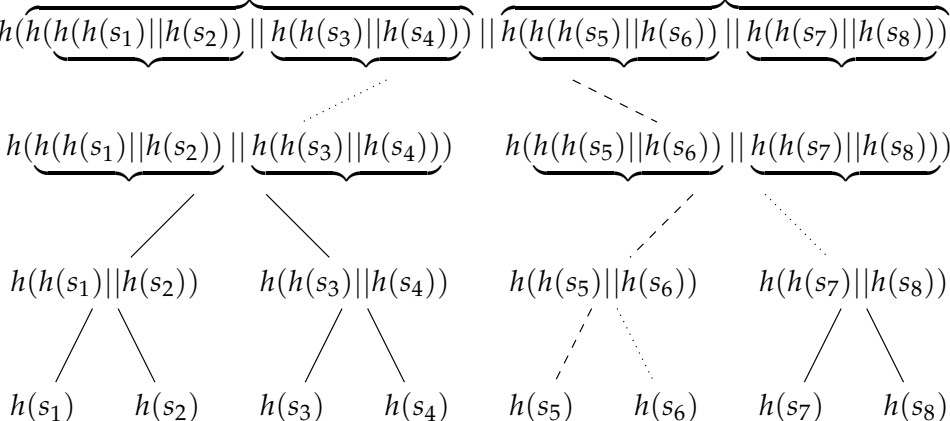

**Figure 8.** A hash tree for data blocks $s_1, \ldots, s_8$. The path from the leaf labeled $h(s_5)$ to the root is shown with dashed lines. Siblings of nodes in that path are connected to their parents by dotted lines.

### 3. Hashcash Trees and Their Application to Client Puzzle Protocols

The definition of hashcash tree is parameterized with respect to a hash function $h$ and a positive integer $k$, which are assumed fixed in this section. Recall that $h_k(s)$ denotes any pair $\langle h(s||x), x \rangle$ such that $x \in \mathbb{N}$ and $\mathit{prefix}(h(s||x), k) = 0^k$. For a string $s$ and an integer $n \in \mathbb{N}^+$, let $(\ell_i)_{i \in \mathbb{N}^+}$ be the (nondeterministic) sequence of labels defined as follows:

$$\ell_i := \begin{cases} h_k(s||i||(\ell_{2i}|_1)||(\ell_{2i+1}|_1)) & \text{if } i \in [1..n]; \\ \langle \epsilon, 0 \rangle & \text{otherwise.} \end{cases} \tag{2}$$

The *hashcash tree* of size $n$ for the string $s$ is the perfect tree $T$ of height $\lceil log_2(n) \rceil$ such that, for each $v \in nodes(T)$, if $order(v, T) = i$, then $label(v, T) = \ell_i$. As shown by Algorithm 3 and Figure 9, a hashcash tree of size $n$ is constructed by first computing the labels of the nodes of level $\lceil log_2(n) \rceil$ (i.e., leaves) and then iteratively computing the labels of nodes of previous levels until the label of the root is obtained. In total, the algorithm performs $n$ hashcash computations.

---

**Algorithm 3:** HASHCASHTREE($s, n, h, k$)

1   $height := \lceil log_2(n) \rceil$;
2   $tree := \textbf{array}[2^{height}]$;            // new array of $2^{height}$ elements
3   **for** $i := 2^{height}$ **down to** 1 **do**
4      **if** $i > n$ **then**
5         $tree[i] := \langle \epsilon, 0 \rangle$;
6      **else**
7         $tree[i] := \text{HASHCASH}(s||i||tree[2i][1]||tree[2i+1][1], h, k)$;

8   **return** $tree$;

---

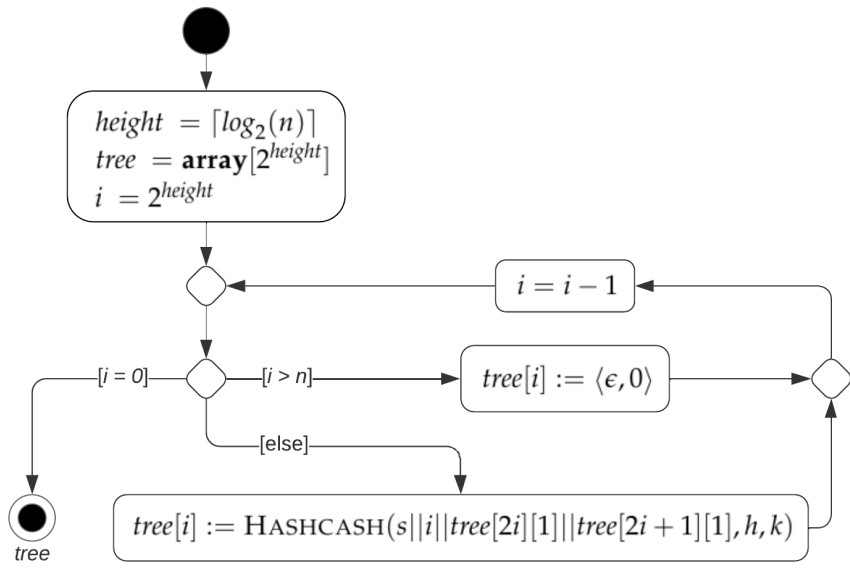

**Figure 9.** Activity diagram of Algorithm 3 building a hashcash tree of size $n$ for the string $s$ using hash function $h$ and prefix length $k$.

**Example 7.** *Let $T$ be the hashcash tree shown in Figure 10. Let $v_0$ denote the node of $T$ such that $order(v_0, T) = 0$. Once the label of the root is disclosed, changing any label in the tree is computationally unfeasible. In particular, changing the label of a leaf, say $v_4$, requires violating second-preimage resistance of $h$ at each level of $T$, with the additional difficulty that the new hash values must also be a valid output of the hashcash algorithm.*

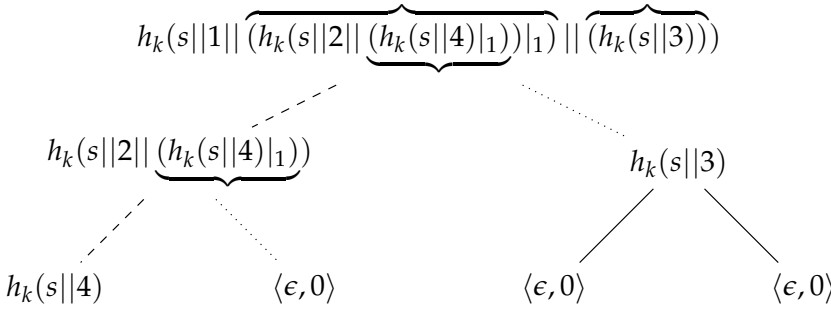

**Figure 10.** A hashcash tree of size 4 for the string *s*. The path from the first leaf (node with order 4) to the root is shown with dashed lines. Siblings of nodes in that path are connected to their parents by dotted lines.

---

**Algorithm 4:** HASHCASHTREEVERIFY($T, i, s, n, h, k$)

// Let $v_o$ be the node of $T$ such that $oder(v_o, T) = o$, for $o \in [1..n]$.

1 **while** $i \geq 1$ **do**
2     **if** $i \leq n$ **then**
3        $left := label(v_{2i}, T)$ **if** $2i \leq n$ **else** $\epsilon$;
4        $right := label(v_{2i+1}, T)$ **if** $2i + 1 \leq n$ **else** $\epsilon$;
5        **if not** HASHCASHVERIFY($label(v_i, T), s||i||left||right, h, k$) **then**
6           **return false**;

7     $i := i$ **div** 2;
8     **if** $prefix(label(v_{2i}, T)|_1) \neq 0^k$ **or** $prefix(label(v_{2i+1}, T)|_1) \neq 0^k$ **then**
9        **return false**;

10 **return true**;

---

Similarly to a hash tree, the validation of a leaf *i* of a hashcash tree *T* involves nodes in the path from *i* to the root of *T* and their children; see Algorithm 4 and Figure 11. Specifically, the prefix of all hash values associated with these nodes is validated (lines 8–9), while hash values are recomputed only for nodes in the path from *i* to the root of *T* (lines 2–6). On the basis of Algorithms 3–4, the proposed CPP comprises the following steps:

1.  The setup consists in the verifier *V* generating and storing a master key *mk*.
2.  The prover *P* needs to send a request *req* to *V*. To this aim, *P* sends $Req := h(req)$ to *V*.
3.  *V* determines the difficulty parameter $n \in \mathbb{N}^+$ based on its current workload, generates a timestamp *t* by which the protocol must be completed, computes $s := h(mk||Req||n||t)$, and sends $\langle s, n, t \rangle$ to *P*.
4.  *P* computes and stores the hashcash tree *T* of size *n* for *s* using Algorithm 3 and sends $\langle sol, s, n, t, Req \rangle$ to *V*, where *sol* is $label(root(T))|_1$.
5.  *V* checks $s = h(mk||Req||n||t)$, verifies that *t* is in the future, randomly selects a number $i \in [2^{H-1}..2^H - 1]$ (a leaf), and sends $\langle I, i \rangle$ to *P*, where $I := h(mk||Req||sol||i)$.
6.  *P* sends $\langle req, S, I, i, s, n, t, Req \rangle$ to *V*, where *S* consists of labels associated with nodes in the path from *i* to the root and their children; in formulas, *S* is the sequence comprising $\ell_{\lfloor i \cdot 2^{-j} \rfloor}$ and $\ell_{\lfloor i \cdot 2^{-j} \rfloor + 1}$, for $j \in [0..\lceil log_2(n) \rceil - 1]$. (As an optimization, witness integers of labels not in the path from *i* to the root can be discarded.)
7.  *V* checks all the following conditions: $s = h(mk||Req||n||t)$; *t* is in the future; $I = h(mk||Req||sol||i)$, where *sol* is $\ell_1|_1$ in *S* (i.e., the hash value associated with the root of the partial hashcash tree sent by *P*); *S* is valid; $Req = h(req)$. If all conditions are met, *req* is processed.

The validation of *S* at step 7 amounts to check that each label $\ell_j$ in the path from *i* to the root is actually obtained according to (2); Algorithm 4 is used.

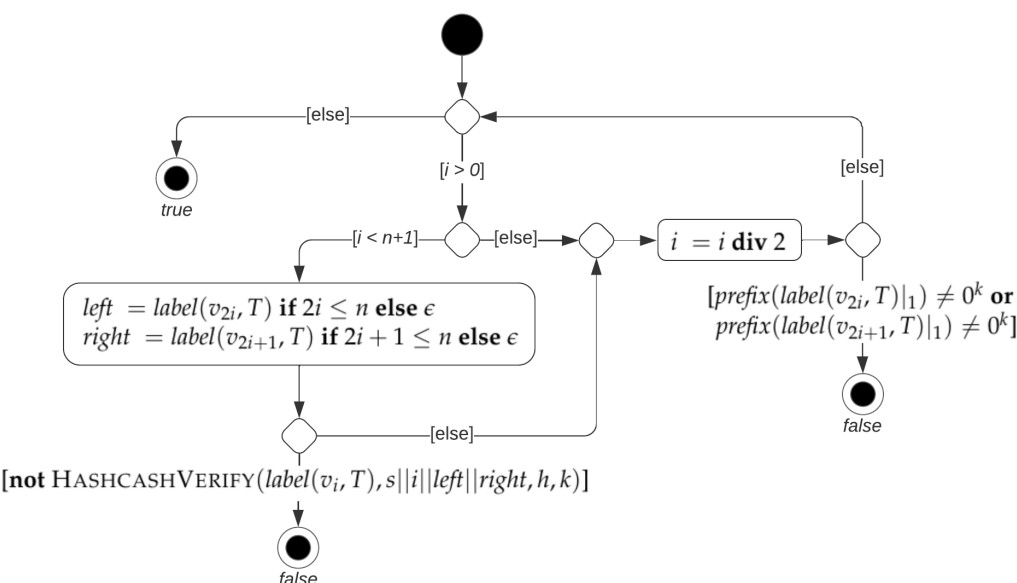

**Figure 11.** Activity diagram of Algorithm 4 verifying node *i* of a hashcash tree *T* of size *n* for the string *s* using hash function *h* and prefix length *k*. Recall that $v_o$ denotes the node of *T* such that $oder(v_o, T) = o$, for $o \in [1..n]$.

**Example 8.** *Let us run the CPP for req = "req", using the hash function SHA-256 and a prefix length $k = 4$. V generates the master key $mk = 41416d572ab944bab51deb6ab388c434$. P sends $Req = c3f7bdf537c46724392c4428e47e04c148c56966190c3c9ed92114800c9f35bb$ to V. V determines $n = 4$ (encoded as 0400), $t = 1691442051.890254$ (10 s in the future, encoded as $ecf9f8e05634d941$), $s = SHA\text{-}256(mk||Req||n||t) = d4b147ccb397af1b3a1f9d278e8 edaba350530291bcc0cf211cafd1042dc1ed6$, and sends $\langle s, n, t \rangle$ to P. P computes the hashcash tree shown in Figure 12, and sends $sol = 002eb1f4d23d95984fa2be7280d4e4fddf65c2f94d757 3b31a426885b115a9e6$ (with $s, n, t$, and Req) to V. V checks that s and t are valid, randomly selects $i = 4$ (the first leaf), computes $I = 8125fc9ab943890cd3d3f5ec0031f8a5fb3fe392b47a59 19d4a06cf4c371da20$, and sends $\langle I, i \rangle$ to P. P sends the labels $\langle sol, 3 \rangle$, $\langle h_2, 13 \rangle$, $\langle h_3, 3 \rangle$, $\langle h_4, 15 \rangle$ and $\langle \epsilon, 0 \rangle$ to V (with the other required data). V runs Algorithm 4 on a tree T constructed with the received labels (other labels are irrelevant). Since the algorithm returns **true**, and all other conditions are met, the request req is processed.*

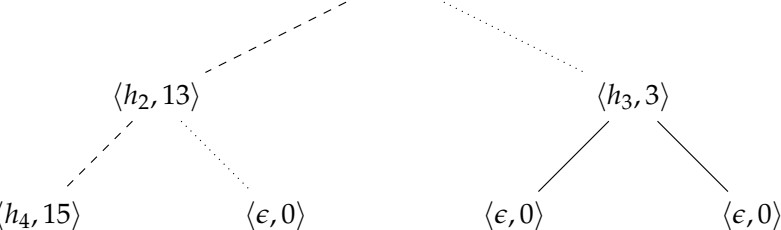

**Figure 12.** A hashcash tree of size 4 for $d4b147ccb397af1b3a1f9d278e8edaba350530291bcc0cf211cafd10 42dc1ed6$, where $h_2 = 08b706ecf8f3a4f73a4adfbbc3be88c39b17e970892d3e41c92935b3354acdd1$, $h_3 = 0e404 64aa8121dc6eaa9d676c9b44d3d7ad4d9a1d48776ecffea0b68b6ac068e$ and $h_4 = 09803620d374bac45cdad30712 578610dae021c2cd394234a2b8749d36fcee11$. The path from the first leaf (node with order 4) to the root is shown with dashed lines. Siblings of nodes in that path are connected to their parents by dotted lines.

## 4. Properties

### 4.1. Computational Complexity

The complexity of the algorithms introduced in the previous section is analyzed by measuring the number of computed hash values. Let *s* be a message and *h* be a crypto-

graphic hash function producing hash values of length $m \in \mathbb{N}$. By definition of cryptographic hash function, the probability that $h(s) = s'$ is $2^{-m}$, for all $s' \in \{0,1\}^m$. Hence, the probability that $h(s)$ starts with the prefix $0^k$ is $2^{-k}$, for all $k \in [0..m]$. Since all hash value computations are independent, the number of hash values computed by Algorithm 1 follows the geometric distribution. As for the verification procedure (Algorithm 2), a single hash value is computed.

**Proposition 1.** *The probability that* HASHCASH$(s, h, k)$ *terminates after computing* $N + 1$ *hash values is* $(1 - 2^{-k})^N \cdot 2^{-k}$, *for all* $N \in \mathbb{N}$. *On average, the number of unused hash values computed by Algorithm 1 is* $(1 - 2^{-k}) \cdot 2^k$, *with variance* $(1 - 2^{-k}) \cdot 2^{2k}$.

**Proof.** Since all hash values are equiprobable by definition of $h$, the probability that 0 occurs as a specific output bit is 0.5. As all bits are independent, the probability that a hash value starts by $0^k$ is $p = 2^{-k}$. The algorithm terminates at the first success of independent trials, so the probability that it terminates after computing $N + 1$ hash value is modeled by the geometric distribution: the probability mass function (of observed failures) is $(1 - p)^N \cdot p$, the mean value (of observed failures) is $(1 - p)/p$, and the variance is $(1 - p)/p^2$. The proof is complete after substituting $p = 2^{-k}$. $\square$

**Proposition 2.** HASHCASHVERIFY$(\langle hash\_value, x \rangle, s, h, k)$ *terminates after computing one hash value.*

**Proof.** Immediate by observing line 1 of Algorithm 2. $\square$

Regarding hashcash trees, the first $n$ nodes of a hashcash tree of size $n$ are labeled by hashcash output values. The prefix $0^k$ is common to all such hashcash output values, hence keeping the success probability constant for all hashcash computations. It turns out that the number of hash values computed by Algorithm 3 follows the negative binomial distribution.

**Theorem 1.** *The probability that* HASHCASHTREE$(s, n, h, k)$ *terminates after computing* $N + n$ *hash values is*

$$\binom{N+n-1}{N}(1 - 2^{-k})^N \cdot 2^{-k \cdot n} \tag{3}$$

*for all* $N \in \mathbb{N}$. *On average, the number of unused hash values computed by Algorithm 3 (via Algorithm 1) is* $n \cdot (1 - 2^{-k}) \cdot 2^k$, *with variance* $n \cdot (1 - 2^{-k}) \cdot 2^{2k}$.

**Proof.** Each call to Algorithm 1 follows the Bernoulli distribution with success probability $p = 2^{-k}$. Therefore, the total number of unused hash values (failures) follows the negative binomial distribution: the probability mass function (of observed failures) is

$$\binom{N+n-1}{N}(1 - p)^N \cdot p^n, \tag{4}$$

the mean value (of observed failures) is $n \cdot (1 - p)/p$, and the variance is $n \cdot (1 - p)/p^2$. The proof is complete after substituting $p = 2^{-k}$. $\square$

Regarding the verification procedure (Algorithm 4), the labels that are verified are those in the path from the selected leaf to the root.

**Theorem 2.** HASHCASHTREEVERIFY$(T, i, s, n, h, k)$ *terminates after computing at most* $\lceil log_2(n) \rceil$ *hash values.*

**Proof.** Hash values are computed indirectly by calling Algorithm 2, one hash value for each call. Algorithm 2 is called at line 5, at all iterations of the main loop in the worst case. The main loop is repeated $\lceil log_2(n) \rceil$ in the worst case because $i$ is divided by 2 at line 7. □

*4.2. Determinable Difficulty*

Theorem 1 provides a clear indication that the average number of hash values computed by Algorithm 3 scales linearly on the size of the hashcash tree, and exponentially on the length of the required prefix. Similarly, and more importantly, the expected variance is linear with respect to the size of the hashcash tree, and exponential with respect to the prefix length. It turns out that, in order to control the average number of hash values with a relatively small variance, Algorithm 3 must be run with small values of $k$, adjusting the size $n$ to impose the difficulty of the problem. Figure 13 reports the average number of hash values and the expected standard deviation for several values of $k$ and $n$.

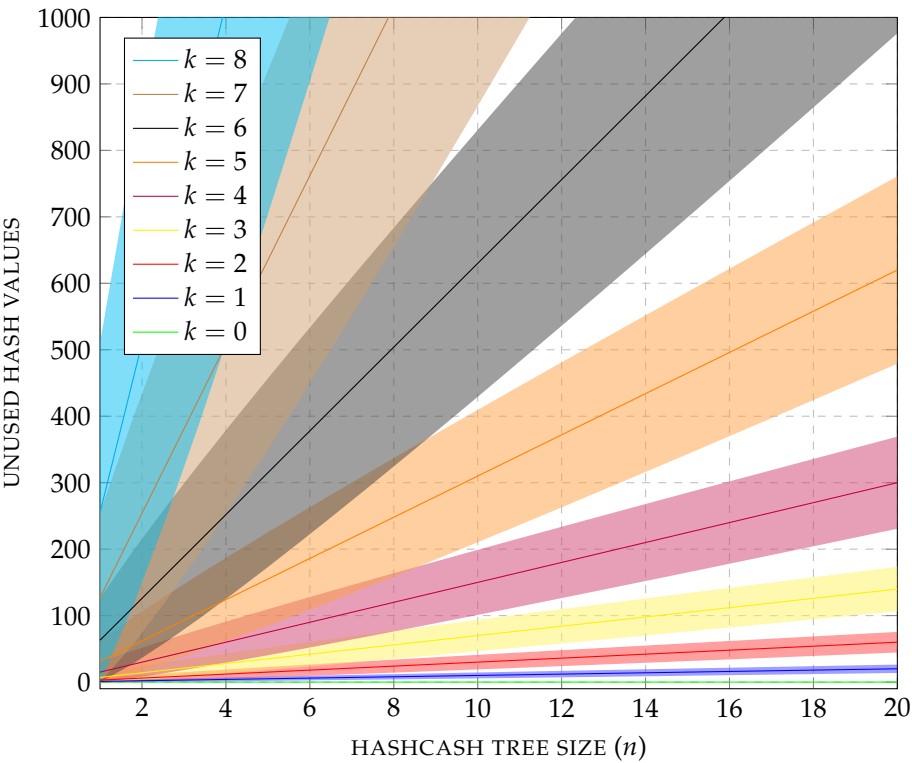

**Figure 13.** Average number of unused hash values computed by HASHCASHTREE$(s, n, h, k)$ for different values of the prefix length $k$. Numbers within the standard deviation are shown by a shade of the same color.

*4.3. Parallel Computation Resistance*

As already observed, a parallel version of Algorithm 1 is expected to have linear speedup. With the proposed protocol, parallel computation resistance is achieved over the paths of the hashcash tree, as in fact the labels of internal nodes are obtained by computing hashcash output values of strings including the hash values associated with child nodes. It turns out that the height of the hashcash tree provides a measure on the parallel computation resistance.

**Theorem 3.** *Any parallel implementation of* HASHCASHTREE$(s, n, h, k)$ *includes at least* $\lceil log_2(n) \rceil$ *sequential calls to Algorithm 1.*

**Proof.** The computation of the label of each internal node in the hashcash tree is a synchronization point: labels of child nodes must be computed before the starting the computation

of the label of the internal node. By definition, there is at least one leaf of the hashcash tree whose label is different from $\langle \epsilon, 0 \rangle$. Hence, all nodes in the path from such a leaf to the root have labels obtained by sequential calls to Algorithm 1. The length of such a sequence is the height $\lceil log_2(n) \rceil$ of the tree. $\square$

## 5. Implementation and Experiment

A proof-of-concept implementation of the algorithm presented in Section 3 is available at https://github.com/alviano/hashcash-tree (accessed on 28 September 2023). It is written in Python 3.11 and uses the SHA-256 function from the `hashlib` package (even if other hash functions can be easily added). Numeric computation is powered by NumPy, and witnesses are represented as unsigned 16-bits integers. The CPP is implemented in a REST server powered by the FastAPI framework. Master keys are *universally unique identifier* (UUID) v4, i.e., 36-character alphanumeric strings. Timestamps are represented as double precision floating-point numbers. An example client for consuming the REST API is also provided. It performs 1000 (one thousand) requests, possibly using multiple threads or processes. Requests themselves are not of particular importance for the example client, which is designed so that all the computation is focused on solving the CPP provided by the REST server. Note that fixing $n = 1$ in the implemented CPP essentially results into the CPP defined in Section 2.2; the prover is challenged to solve a single hashcash computation, for some prefix length $k$. On the other hand, fixing $k = 0$ in the implemented CPP essentially results in a CPP using hash trees because hashcash is disabled.

In order to empirically verify the theoretical analysis carried out in Section 4, the REST server and the example client were run with several configurations. The difficulty of the generated puzzles was varied by modifying both the prefix length (parameter $k$) and the size of the hashcash tree (parameter $n$); within this respect, $k$ was tested with all values from 0 to 8, and $n$ was tested with value $2^i - 1$ for $i \in [4..15]$. As for the client, since the experiment was run on a quad-core Intel(R) Xeon(R) CPU X3430 @ 2.40 GHz with 16 GB of RAM, the number of workers was fixed to 4 (i.e., the 1000 requests were processed in parallel by 4 processes). Measured values include the total CPU usage of the REST server, and the CPU usage of the example client for each request (from starting the interaction with the REST server to the submission of the validation data). Computed values include the average CPU usage for completing the 1000 requests, the standard deviation, and the minimum and maximum CPU usage.

A summary of the measured and computed values is shown in Figure 14. The plot uses logarithmic axes and reports the average CPU usage of the prover; times within the standard deviation are colored in dark shades, and other times between the minimum and the maximum CPU usage are colored in light shades. As a first observation, each increment of the prefix length ($k$) causes a jump in the effort required by the prover to solve the puzzle. Recall that for $n = 1$, the CPP is essentially the one based on hashcash alone. On the other hand, for every fixed value of $k$, the prover effort scales linearly on the size of the hashcash tree ($n$). Observing the standard deviation, it is possible to conclude that all hashcash trees are computed with similar effort once $k$ and $n$ are fixed. As a final observation on the plot, note that for $k = 0$ (i.e., essentially using hash trees) the puzzle is solved in less than 2 s even for the largest case of $n = 2^{15} = 32{,}767$. If storing a node takes 34 bytes (32 bytes for the SHA-256 hash value and 2 bytes for the witness), a hashcash tree of size $n = 2^{15} = 32{,}767$ requires around 1024 KiB of memory. In contrast, note that for $k = 4$ and $n = 4095$ the puzzle is solved in around 2.26 s and the hashcash tree can be stored in around 136 KiB. Similarly, for $k = 5$ and $n = 2047$ the puzzle is solved in around 2.20 s and the hashcash tree can be stored in around 68 KiB.

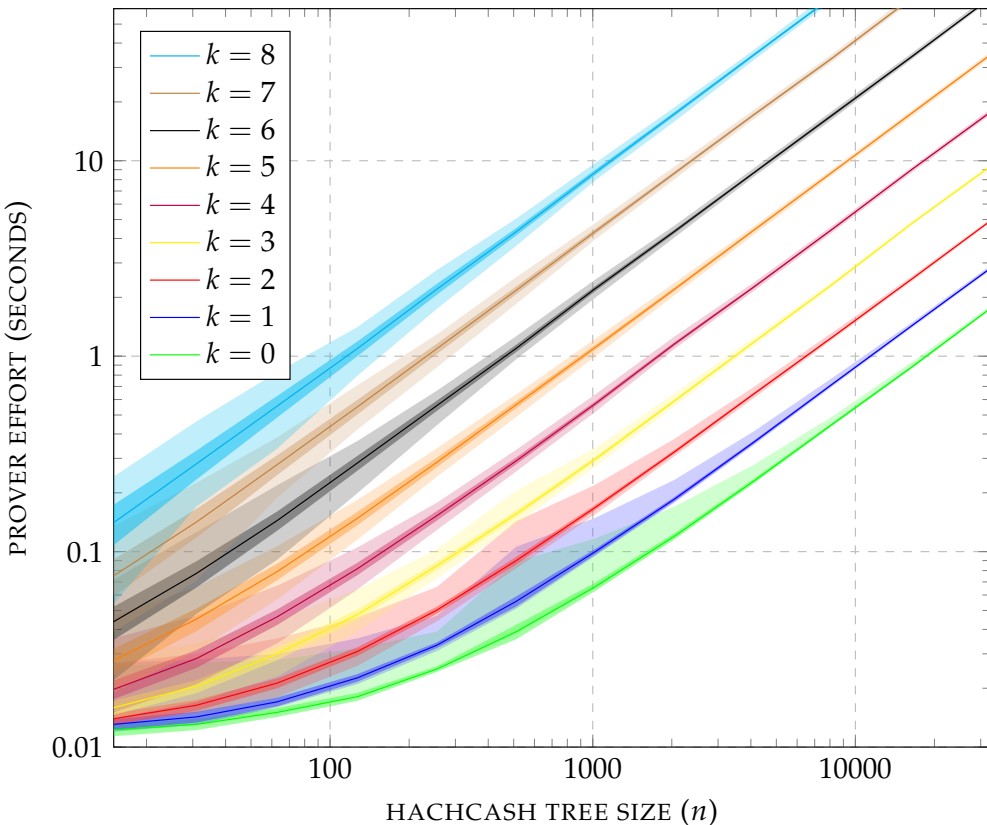

**Figure 14.** Average prover effort (*P*'s CPU time, in seconds) over 1000 requests, with standard deviation (dark shade), and minimum and maximum values (light shade). For each prefix length (parameter *k*), the size of the hashcash tree varies from $n = 2^4 - 1 = 15$ to $n = 2^{15} - 1 = 32{,}767$.

Figure 15 shows the measured prover effort for each solved CPP. There is a plot for each tested value of *n*. Each of these plots reports one line for each tested value of *k*. The lines are obtained by plotting the measured CPU time (*y* axis) for each solved CPP (*x* axis). It can be observed that the computation of hashcash trees of size up to $n = 63$ is very fast, always below 1 s. On the contrary, the computation of hashcash trees of size $n = 8191$ is very slow for $k \geq 5$, requiring at least 10 s. Focusing on the remaining values of *n*, from 127 to 4095, the values of *k* that lead to CPU times between 0.1 s and 10 s are 4, 5, and 6.

Figure 16 is focused on the values of *k* identified above. The benchmark was run by increasing the size of the generated hashcash tree linearly, with steps of 32 nodes. For each tested size, 1000 hashcash trees were generated. The plot reports the average CPU time used by the prover, with values within the standard deviation and within the minimum and maximum measured values. For all three prefix lengths, the prover effort scales linearly, confirming that the verifier can precisely control the difficulty of the puzzle. The measured verifier effort, including the CPU usage for running FastAPI, is the following: for $k = 4$, it is around 12.7 ms per request, with standard deviation 0.3 ms; for $k = 5$, it is around 12.9 ms per request, with standard deviation 0.4 ms; for $k = 6$, it is around 13.4 ms per request, with standard deviation 0.6 ms.

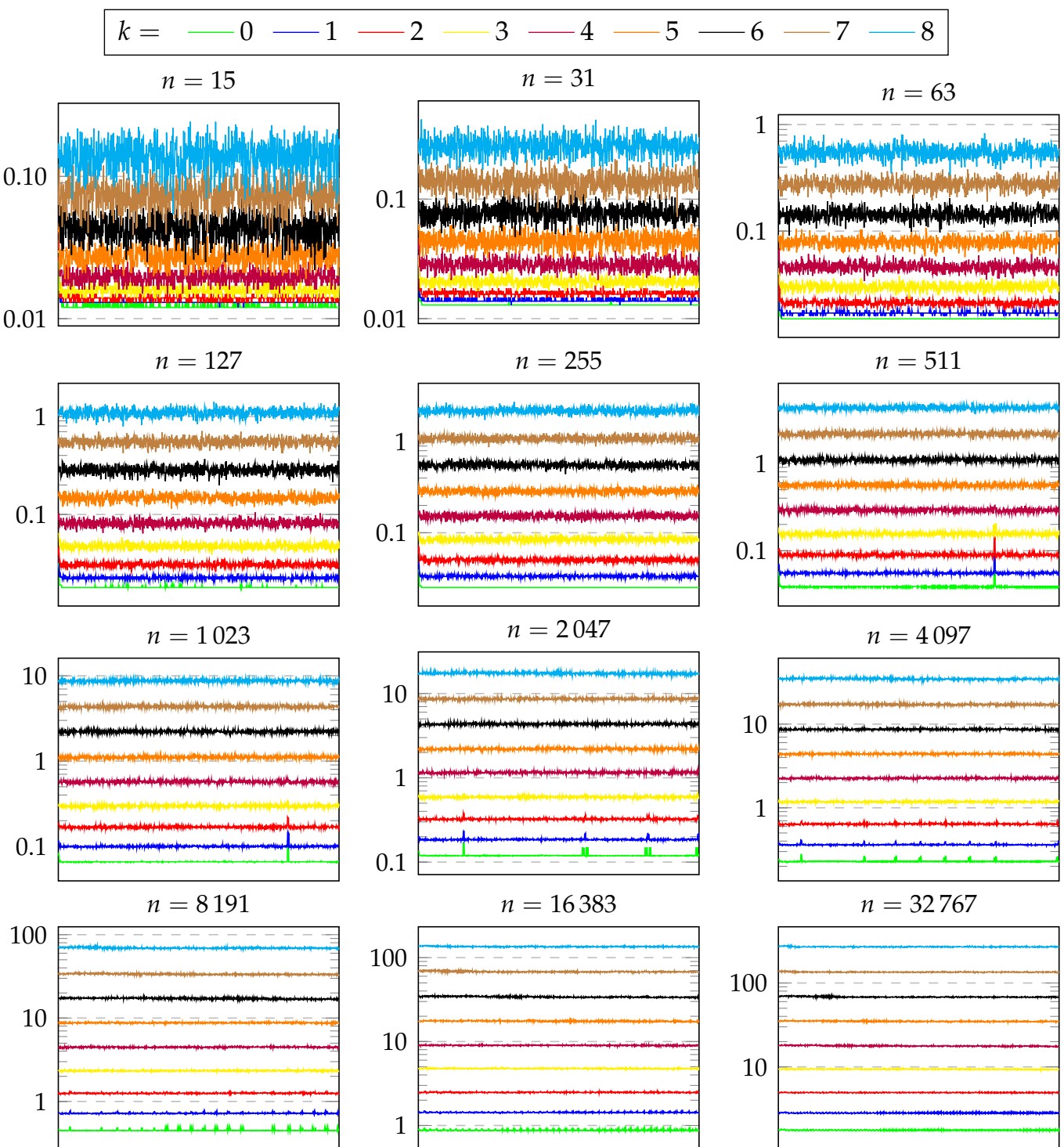

**Figure 15.** Prover effort (*P*'s CPU time, in seconds) over 1000 requests for different prefix lengths (parameter *k*) and size of the hashcash tree (parameter *n*).

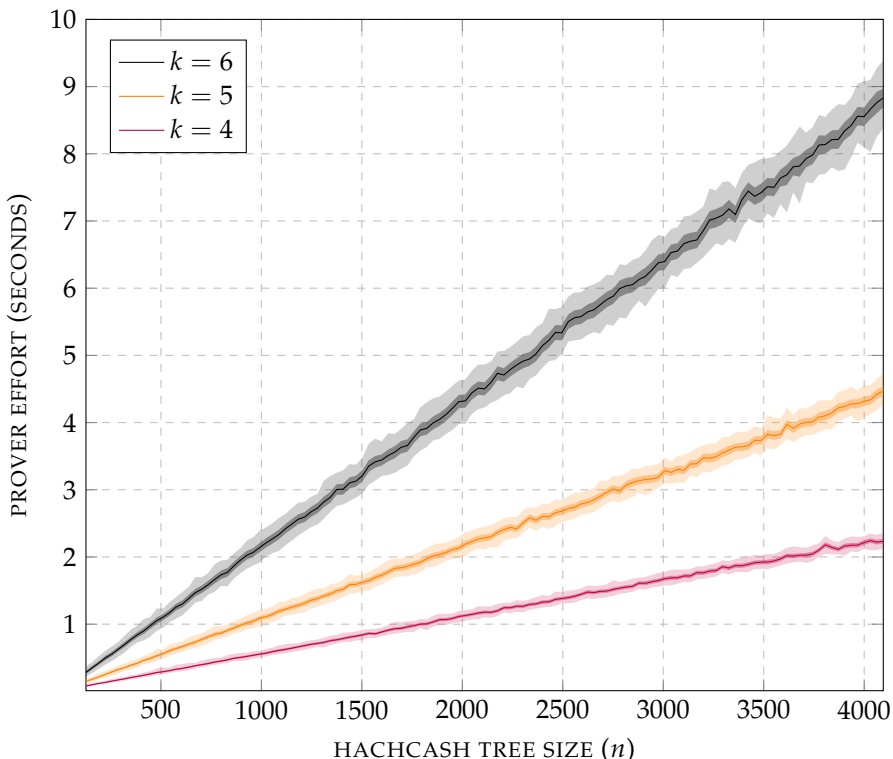

**Figure 16.** Average prover effort (*P*'s CPU time, in seconds) over 1000 requests, with standard deviation (dark shade), and minimum and maximum values (light shade). Prefix length (parameter *k*) fixed to 4, size of hashcash tree varying from $n = 127$ to $n = 4095$ with steps of 32.

## 6. Related Work

Detecting DoS attacks is challenging and addressed by sophisticated techniques, among them some based on machine learning [4–6]; a survey on DoS detection techniques is given by de Neira et al. [7]. As an earlier barrier against DoS attacks, prevention techniques can be adopted to protect sensitive services and assets of an organization [14,15]. This article introduces a client puzzle protocol as a prevention technique to mitigate DoS attacks.

The concept of the client puzzle was introduced by Juels and Brainard [16], who suggested their application to prevent denial-of-service (DoS) attacks. The main characteristic of client puzzles is that they can be solved by a polynomial-time entity upon spending a certain amount of resources, and therefore a server may provide access to some of its assets in exchange of a valid solution for a new client puzzle. A similar concept is given by Dwork and Naor [9] with the notion of pricing function to combat junk emails, and by Rivest, Shamir, and Wagner [17] with the notion of timed-lock puzzle as a tool to realize timed-release crypto. Client puzzles are expected to be unforgeable and difficult to solve [18] and possibly to have determinable difficulty and parallel computation resistance [19].

Client puzzles can be categorized as CPU-bound and memory-bound. In CPU-bound client puzzles, the prover effort is measured by the amount of CPU cycles needed to solve a puzzle; several client puzzles belong to this category [9,11,16–18,20–22]. In memory-bound client puzzles, the prover effort is measured by the amount of memory look-ups needed to solve a puzzle; the main argument in support of memory-bound client puzzles is that CPU power varies more than memory look-up speed for different computers [23–25].

The CPP presented in Section 3 is CPU-bound and combines hashcash [11] and hash trees [26]. The main obstacle to using hashcash alone is its unbounded probability cost. The length *k* of the prefix is the only parameter that can be used to control the difficulty of the puzzle, and both difficulty and variance increase exponentially when *k* increases (see Proposition 1). An attempt to gain more control on the difficulty of puzzles was performed by Juels and Brainard [16], who essentially designed a CPP involving several sub-puzzles.

The results shown by Theorem 1 for hashcash trees can be extended to such a CPP when sub-puzzles are hashcash computations. On the other hand, such a CPP requires verifying (and therefore transmitting) the solutions of all sub-puzzles, while a logarithmic number of solutions is sufficient to verify a hashcash tree (see Theorem 2).

A CPP based on hash trees was designed by Coelho [12]. It can be seen as a solution–verification version of the CPP proposed in Section 3; when the prefix length is fixed to $k = 0$ (i.e., hashcash is disabled), the tree is computed based on the service description and several leaves are selected for the verification phase based on the root hash value. The difficulty of the puzzle is determinable with high precision, but the size of hash trees can grow quickly. This is a downside of the protocol, given the fact that the hash tree must be stored (or recomputed) by the prover in order to provide the labels for the verification phase, which are discovered only after the root hash value is computed. The proposed CPP can rely on smaller trees because the difficulty of computing a single node can also be controlled via the length of the required prefix, i.e., by enabling hashcash.

Differently from previously defined protocols, in addition to the first interaction with the server to obtain the challenge for accessing the requested service, the proposed protocol expects a commitment on the computed solution before disclosing the portion required to prove the legitimacy of the client (see Figure 4). This is in particular contrast with the non-interactive approach by Raikwar and Gligoroski [27], whose protocol is explicitly designed to limit the interaction with the server to the verification phase (see Figure 2). Another fundamental difference with the protocol by Raikwar and Gligoroski is the adopted cryptographic technique: Raikwar and Gligoroski opted for deterministic *verifiable delay function* (VDF) [28], while the protocol proposed in this article is based on the non-deterministic hashcash algorithm. Given the fact that the non-determinism is essentially mitigated by the use of short prefixes, as shown in Section 5, adopting hashcash instead of VDF is justified by a simpler implementation.

## 7. Conclusions

Hashcash trees combine features of the hashcash algorithm with those of hash trees. Labels are obtained by running the hashcash algorithm and therefore are moderately hard to compute (exponential on the length of the prefix) and easy to verify (one hash value computation). Labels of internal nodes depend on child nodes, and therefore parallel computation is limited. Moreover, the root is a commitment for the tree, which does not need to be fully transmitted to the verifier. In fact, the verification involves a logarithmic number of nodes because the selected leaf is known to the prover only after disclosing the commitment. The associated client puzzle protocol relies on two parameters for controlling the prover's effort. The number of computed hash values grows exponentially on the prefix length and linearly on the size of the hashcash tree. The empirical analysis suggests that the prefix length can be fixed to 4 or 5, with hashcash trees of size between 127 and 4095, to generate puzzles solvable in a few seconds. Improving parallel computation resistance is an interesting future line of research and will require the introduction of synchronization points in the computation of labels at the same level of the tree. Other future lines of research include the definition of a (one-phase) challenge–response protocol, for example, by self-imposing the selected leaf based on the hash value of the root node, and the definition of a solution–verification protocol, constructing the hashcash tree based on publicly-available data from the server.

**Funding:** This research was partially funded by Italian Ministry of Research (MUR) under PNRR project FAIR "Future AI Research", CUP H23C22000860006, under PNRR project Tech4You "Technologies for climate change adaptation and quality of life improvement", CUP H23C22000370006, and under PNRR project SERICS "SEcurity and RIghts in the CyberSpace", CUP H73C22000880001; by Italian Ministry of Health (MSAL) under POS project RADIOAMICA, CUP H53C22000650006; by the LAIA lab (part of the SILA labs), and by GNCS-INdAM.

**Data Availability Statement:** No new data were created or analyzed in this study. Data sharing is not applicable to this article.

**Conflicts of Interest:** The author declares no conflict of interest. The funders had no role in the design of the study; in the collection, analyses, or interpretation of data; in the writing of the manuscript; or in the decision to publish the results.

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
