# Peer review of "Hashcash Tree, a Data Structure to Mitigate Denial-of-Service Attacks"

_algorithms, doi:10.3390/a16100462_

Round 1

Reviewer 1 Report

The authors presented a paper on Hashcash Tree, a data structure designed to mitigate Denial of Service attacks. The paper is scientifically sound and appropriately structured for publication. It includes required sections such as an abstract, keywords, introduction, background, relevant applications, implementation and experiments, related work, conclusion, and references.

Please place the related work section right after the background and present your research questions or list of contributions too. It is not clear what you aim to achieve with these experiments, and it is unclear how this publication will improve upon the current literature. The list of references is rather small; we would expect at least 25-30 references for a journal. Additionally, some of the references are quite old, dating back to 1996, 2004, and 2007. Please also provide a brief discussion of the experiment results in the abstract section.

Author Response

| The authors presented a paper on Hashcash Tree, a data structure designed to mitigate Denial of Service attacks. The paper is scientifically sound and appropriately structured for publication. It includes required sections such as an abstract, keywords, introduction, background, relevant applications, implementation and experiments, related work, conclusion, and references.

Many thanks for your time. I revised the manuscript according to reviewers' suggestions. Most relevant new content is in blue.

| Please place the related work section right after the background and present your research questions or list of contributions too. It is not clear what you aim to achieve with these experiments, and it is unclear how this publication will improve upon the current literature. 

I expanded the introduction as suggested.

I cannot move RW before the technical part because it refers the new contribute.

| The list of references is rather small; we would expect at least 25-30 references for a journal. Additionally, some of the references are quite old, dating back to 1996, 2004, and 2007. 

I added more references.

| Please also provide a brief discussion of the experiment results in the abstract section.

I mentioned the experiment in the abstract.

Reviewer 2 Report

The experiment validated the theoretical analysis and demonstrated the viability of using hashcash trees in client riddle protocols to defend against denial-of-service attacks. The findings provided valuable information regarding the computational complexity and scalability of the protocol, which can be applied to the design and implementation of secure systems.

The experiment demonstrated that increasing the prefix length (k) in the hashcash tree-based client puzzle protocol resulted in a higher effort required by the prover to solve the puzzle. This observation is supported by the finding that the prover effort scaled linearly with the size of the hashcash tree. 

The empirical evaluation showed that hashcash trees of size up to n = 63 could be computed with similar effort once k and n were fixed. This suggests that the computation of hashcash trees within this size range was efficient and feasible for practical implementation.

The standard deviation analysis indicated that hashcash trees were computed with similar effort once k and n were fixed. This suggests that the variability in the prover effort was relatively low, further supporting the efficiency and consistency of the hashcash tree-based client puzzle protocol. 

Despite the observations and questions presented, the work and experiments performed do not diminish in any way. Good article, and it opens up opportunities for others to continue the experiments.

Author Response

Many thanks for your time. I revised the manuscript according to reviewers' suggestions. Most relevant new content is in blue.

| Question 1: How does the performance and scalability of the hashcash tree-based client puzzle protocol compare to other existing client puzzle protocols? Are there any specific advantages or disadvantages of using hashcash trees in terms of computational complexity and resource requirements? 

Added some comments on the experiment. In particular, the CPP based on hashcash alone is essentially obtained for $n=1$, and a CPP relying on hash trees (instead of hashcash trees) is obtained for $k=0$.

| Question 2: Are there any potential trade-offs or limitations associated with increasing the size of the hashcash tree (n) in terms of memory usage or computational overhead? How does the choice of n impact the overall performance and security of the protocol?

Added some comments on the experiment. In particular, regarding the memory required to store hashcash trees: keeping fixed the amount of time, increasing $k$ significantly decreases the amount of memory usage (which depends on n).

| Question 3: Can the hashcash tree-based client puzzle protocol be further optimized or enhanced to improve its efficiency or resistance against potential attacks? Are there any specific areas of improvement or future research directions identified based on the findings of the experiment?

Potential improvements are discussed in the conclusion as future lines of research.

Reviewer 3 Report

This paper proposes a new data structure combining hashcash and Merkle trees. In the proposed data structure, all hash values are required to start with a given number of zeros, and hash values of internal nodes are obtained by hashing the hash values of child nodes. The client is forced to compute all hash values, but only those in the path from a leaf to the root are required by the server to verify. This paper presents a highly promising solution with certain potential for practical applications in mitigating DoS attacks.

While the manuscript showcases several strengths, there are some problems which must be solved before it is considered for publication. 

1. The Introduction section is unclear in its explanation of the paper's contributions. While the introduction provides a reasonable overview of the research context and mentions the compared algorithms like POW (introduced in 1992) and Coelho's solution-verification protocol based on hash trees (from 2008), it lacks a clear and concise articulation of the novelty and advantages of the proposed method. To improve the clarity of the Introduction section, I recommend adding a dedicated paragraph that explicitly outlines the innovative aspects and strengths of your method, and providing more relevant information regarding recent research to enhance the credibility of the paper.

2.The Background introduces numerous concepts and provides illustrative examples. However, this section lacks a concise summary or overarching explanation of its content, as well as a clear relevance between the introduced concepts and the method in this paper.

3.The content of Algorithm 1 and Algorithm 2 is too simple, perhaps it could be explained using equations instead of pseudocode.

4.In Hashcash Trees and Their Application to Client Puzzle Protocols, the authors have provided a comprehensive explanation of the crucial steps involved in their proposed CPP implementation. However, I recommend adding flowchart to explain the algorithm. This addition will not only enhance the comprehensibility of the paper but also contribute to the overall quality of the manuscript.

5.The Related Work is lacking in recent research analysis and comparisons, and the number of cited works is relatively limited. This deficiency makes it challenging for readers to gain a comprehensive understanding of the research background and related work. I encourage the authors to review and include the most relevant and recent studies in the Related Work section to enhance the quality and comprehensiveness of this part of the manuscript.

In conclusion, your manuscript presents a highly promising solution with significant potential for addressing DoS attacks, and it deserves recognition for its contributions to network security. I encourage you to address the aforementioned issues during the revision process to enhance the overall quality and readability of the paper.

The language and writing style of the manuscript require further refinement and proofreading.

Author Response

| This paper proposes a new data structure combining hashcash and Merkle trees. In the proposed data structure, all hash values are required to start with a given number of zeros, and hash values of internal nodes are obtained by hashing the hash values of child nodes. The client is forced to compute all hash values, but only those in the path from a leaf to the root are required by the server to verify. This paper presents a highly promising solution with certain potential for practical applications in mitigating DoS attacks.

Many thanks for your time. I revised the manuscript according to reviewers' suggestions. Most relevant new content is in blue.

| While the manuscript showcases several strengths, there are some problems which must be solved before it is considered for publication. 

| 1. The Introduction section is unclear in its explanation of the paper's contributions. While the introduction provides a reasonable overview of the research context and mentions the compared algorithms like POW (introduced in 1992) and Coelho's solution-verification protocol based on hash trees (from 2008), it lacks a clear and concise articulation of the novelty and advantages of the proposed method. To improve the clarity of the Introduction section, I recommend adding a dedicated paragraph that explicitly outlines the innovative aspects and strengths of your method, and providing more relevant information regarding recent research to enhance the credibility of the paper.

Introduction expanded as suggested.

| 2.The Background introduces numerous concepts and provides illustrative examples. However, this section lacks a concise summary or overarching explanation of its content, as well as a clear relevance between the introduced concepts and the method in this paper.

Added subsections and a summary to anticipate the content of the background section.

| 3.The content of Algorithm 1 and Algorithm 2 is too simple, perhaps it could be explained using equations instead of pseudocode.

Kept algorithms for uniformity with the Algorithms 3 and 4.

| 4.In Hashcash Trees and Their Application to Client Puzzle Protocols, the authors have provided a comprehensive explanation of the crucial steps involved in their proposed CPP implementation. However, I recommend adding flowchart to explain the algorithm. This addition will not only enhance the comprehensibility of the paper but also contribute to the overall quality of the manuscript.

Added activity diagrams for Algorithms 3 and 4.

| 5.The Related Work is lacking in recent research analysis and comparisons, and the number of cited works is relatively limited. This deficiency makes it challenging for readers to gain a comprehensive understanding of the research background and related work. I encourage the authors to review and include the most relevant and recent studies in the Related Work section to enhance the quality and comprehensiveness of this part of the manuscript.

Expanded.

| In conclusion, your manuscript presents a highly promising solution with significant potential for addressing DoS attacks, and it deserves recognition for its contributions to network security. I encourage you to address the aforementioned issues during the revision process to enhance the overall quality and readability of the paper.

Reviewer 4 Report

The paper discusses client puzzle protocols used to defend against resource exhaustion denial of service (DoS) attacks. These protocols often involve cryptographic challenges, like finding hash values with specific properties. However, due to how hash functions are designed, it's challenging to predict the difficulty of finding such hash values, a limitation in simple proof-of-work (PoW) algorithms like hashcash. The authors introduce a new data structure, the "hashcash tree," which combines hashcash and Merkle trees. The paper presents empirical evidence showing that this approach allows for accurate control of puzzle difficulty. 

I believe the work makes a good contribution to the field, providing well-documented results that advance our understanding. The methodology is rigorous, and the findings are clearly presented. A valuable addition to the scientific literature.

Author Response

Many thanks.

Reviewer 5 Report

This paper proposes a new data structure to mitigate DoS attacks based on hashcash tree. The proposed client puzzle is implemented and evaluated empirically to show that the difficulty of puzzles can be accurately controlled.

The paper is well-written. The presentation is clear and easy to follow. The technical contribution is valid. The review has no more comments on the revised version.

Minor editing of English language required

Author Response

Many thanks.

Reviewer 6 Report

In this paper, the authors propose a new data structure combining hashcash and Merkle trees,

also known as hash trees. In my opinion, this paper contains a lot of advantages and is well organized, however, it has some major limitations and needs to be modified before being accepted.

(1) The quality of Figures 1 and 3 needs to be further improved. (2) The codes for Examples 2 and 3 should appear as figures. (3) The formulas in the full text should be uniformly numbered. (4) The discussion or comparisons with more recent related schemes, such as tfl-dt: a trust evaluation scheme for federated learning in digital twin for mobile networks, a game theory-based incentive mechanism for collaborative security of federated learning in energy blockchain environment, and anti-tampering scheme of evidence transfer information in judicial system based on blockchain, instead of conventional schemes are suggested.

Good

Author Response

(1) The quality of Figures 1 and 3 needs to be further improved.

Enlarged.

(2) The codes for Examples 2 and 3 should appear as figures. 

Done.

(3) The formulas in the full text should be uniformly numbered. 

Done.

(4) The discussion or comparisons with more recent related schemes, such as tfl-dt: a trust evaluation scheme for federated learning in digital twin for mobile networks, a game theory-based incentive mechanism for collaborative security of federated learning in energy blockchain environment, and anti-tampering scheme of evidence transfer information in judicial system based on blockchain, instead of conventional schemes are suggested.

Expanded RW (but not using the above references, as they are not related with DoS mitigation, and from what I understood the journal does not allow for unrelated citations).

Round 2

Reviewer 3 Report

This paper proposes a new data structure combining hashcash and Merkle trees. However, the comparative analysis of related work in the paper is outdated, lacking recent comparisons with relevant studies from recent years, which fails to demonstrate the novelty of this paper. If indeed there are no recent related works, it is necessary to provide a clear explanation for this. The author's response does not effectively address the previously raised questions. In a whole, the paper cannot be accepted due to lack of novelty.

Moderate editing of English language required.

Author Response

Expanded RW.

Please, note that not all scientific topics are eclectic and have thousands of papers per month. If you have some specific reference to suggest, please provide them to me.

Please, also clarify what is the correct range of years to consider. Is 10 years ago too old?

Reviewer 6 Report

In my opinion, this paper has the following obvious problems, (1) several figures are of poor quality. (2) Many parameters in formulas fail to clearly introduce the meaning and motivation of formula design. (3) The quality of reference literature is not high and the comparison literatures are missing , and the experiment is not convincing. Authors need to improve their papers well and resubmit them for review.

It should be improved.

Author Response

There is nothing that can be used from your review.